# Comparative Study of the Protective and Neurotrophic Effects of Neuronal and Glial Progenitor Cells-Derived Conditioned Media in a Model of Glutamate Toxicity In Vitro

**DOI:** 10.3390/biom13121784

**Published:** 2023-12-13

**Authors:** Georgy Leonov, Diana Salikhova, Margarita Shedenkova, Tatiana Bukharova, Timur Fatkhudinov, Dmitry Goldshtein

**Affiliations:** 1Research Centre for Medical Genetics, 115522 Moscow, Russia; diana_salikhova@bk.ru (D.S.); margarita.shedenkova@yandex.ru (M.S.); bukharova-rmt@yandex.ru (T.B.); dvgoldrm7@gmail.com (D.G.); 2Orekhovich Institute of Biomedical Chemistry of the Russian Academy of Sciences, 119121 Moscow, Russia; 3Research Institute of Molecular and Cellular Medicine, Medical Institute RUDN, 117198 Moscow, Russia; tfat@yandex.ru

**Keywords:** glutamate toxicity, oxidative stress, induced pluripotent stem cells (iPSCs), glial progenitor cells, neuronal progenitor cells, PC12 cells, neurite outgrowth, conditioned medium

## Abstract

Cell therapy represents a promising approach to the treatment of neurological diseases, offering potential benefits not only by cell replacement but also through paracrine secretory activities. However, this approach includes a number of limiting factors, primarily related to safety. The use of conditioned stem cell media can serve as an equivalent to cell therapy while avoiding its disadvantages. The present study was a comparative investigation of the antioxidant, neuroprotective and neurotrophic effects of conditioned media obtained from neuronal and glial progenitor cells (NPC-CM and GPC-CM) on the PC12 cell line in vitro. Neuronal and glial progenitor cells were obtained from iPSCs by directed differentiation using small molecules. GPC-CM reduced apoptosis, ROS levels and increased viability, expressions of the antioxidant response genes *HMOX1* and *NFE2L2* in a model of glutamate-induced oxidative stress. The neurotrophic effect was evidenced by a change in the morphology of pheochromocytoma cells to a neuron-like phenotype. Moreover, neurite outgrowth, expression of *GAP43*, *TUBB3*, *MAP2*, *SYN1* genes and increased levels of the corresponding MAP2 and TUBB3 proteins. Treatment with NPC-CM showed moderate antiapoptotic effects and improved cell viability. This study demonstrated the potential application of CM in the field of regenerative medicine.

## 1. Introduction

The brain exhibits limited regenerative potential, and despite the presence of neural stem cell proliferation in adulthood, the complete replacement of damaged neural tissue remains unattainable [1,2,3]. This limitation complicates the therapeutic management of neurological disorders like ischemic stroke, spinal injuries, and progressive neurodegenerative diseases [4,5]. Current therapies for these pathologies primarily alleviate symptoms and improve patients’ quality of life, but do not facilitate the recovery of impaired functions [6,7]. Parkinson’s disease is conventionally treated with drugs to restore dopamine levels, while Alzheimer’s disease involves the use of selective NMDA receptor inhibitors, and ischemic stroke treatment relies on thrombolysis, albeit with a limited therapeutic window [8,9,10]. Therefore, novel therapeutic strategies for diseases of the central nervous system are needed. Stem cell therapy is promising, especially with the advent of induced pluripotent stem cell (iPSC) technology and approaches to differentiate these cells into various tissue types, including neurons and glial cells [11]. This advance has resolved the ethical concerns associated with embryonic stem cell derivation and enabled the generation of patient-specific pluripotent cells for autologous transplantation [12]. Numerous studies have demonstrated the efficacy of cell therapy by intra-arterial, intravenous or intracerebral stem cell administration [13,14]. The therapeutic benefits of this approach are largely attributed to the paracrine effects of transplanted stem cells [15]. However, this treatment modality is associated with challenges such as immune reactions, oncological transformation and high costs [16]. To overcome these limitations, the use of stem cell conditioned media (CM), instead of direct stem cell implantation, offers a promising alternative. Stem cell CM contain growth factors, neurotrophins and exosomes with therapeutic potential, with most studies focusing on mesenchymal stem cell (MSC) conditioned media due to their accessibility [17,18]. However, the role of factors secreted by glial and neuronal progenitor cells in restoring brain function in neurological diseases remains unclear [19].

Oxidative stress, resulting from hypoxia in ischemic injury and the accumulation of pathological proteins and mitochondrial dysfunction in neurodegenerative diseases, is the central factor in the pathogenesis of these disorders [20]. Neurons, in particular, are highly sensitive to direct oxidative damage, which is a signature of diseases such as Parkinson’s, Alzheimer’s, Huntington’s and amyotrophic lateral sclerosis [21,22]. The intensive metabolism and limited antioxidant capacity of neurons predispose them to the excessive production of reactive oxygen species (ROS) [23]. High levels of extracellular glutamate promote decreased transport of cysteine into the cell via the cystine/glutumate antiporter (Xc system) and a decreased ability of the cell to synthesize the important antioxidant glutathione, leading to increased levels of ROS [24].

The PC12 pheochromocytoma cell line of neoplastic chromaffin cells are widely used to study neurotoxicity and to screen new anti-inflammatory, antioxidant and neuroprotective medicines [25,26]. The key attribute of PC12 as a neuronal cell line is the ability to form neurite outgrowths during differentiation. The neuritogenesis of these cells is induced by many substances, for example, nerve growth factor (NGF), which induces the significant outgrowth of processes. Various models in vitro, such as hypoxia, oxidative stress and mitochondrial dysfunction, are also sufficiently reproduced on this type of cell [27,28,29,30].

The study investigated the neuroprotective, neurotrophic and antioxidant properties of conditioned media derived from neuronal and glial progenitor cells obtained by directed differentiation of iPSCs. The conditioned media were used in a model of oxidative stress induced by glutamate treatment in the PC12 cell line.

## 2. Materials and Methods

### 2.1. Preparation of Neuronal and Glial Progenitor Cells-Derived Conditioned Media

Neuronal and glial progenitor cells (NPCs and GPCs) were obtained previously by directed differentiation iPSCs [31]. NPCs were cultured in Dulbecco’s Modified Eagle Medium/Nutrient Mixture F-12 (DMEM/F12, Gibco, Carlsbad, CA, USA) with 2% B27, 20 ng/mL fibroblast growth factors 2 (FGF-2, ProSpec, Ness-Ziona, Israel) and 2 μM purmorphamine (Stemcell Technologies, Seattle, WA, USA). GPCs were cultured in DMEM/F12 medium, 1% supplement N2 (PanEco, Russia), 2 mM L-glutamine (PanEco, Russia), 100 mg/L penicillin-streptomycin (PanEco, Russia), 1% fetal bovine serum (FBS, Gibco, Carlsbad, CA, USA), 10 ng/mL fibroblast growth factor-2 (FGF-2), 20 ng/mL epidermal growth factor (EGF) and 20 ng/mL ciliary neurotrophic factor (CNTF, ProSpec, Ness-Ziona, Israel) at 37 °C in a 95% humidified incubator with 5% CO_2_. NPCs and GPCs were washed twice with phosphate-buffered saline (PBS) and cultured in DMEM/F12 (1:1) with the addition of 100 mg/L penicillin-streptomycin (PanEco, Russia) during the 16 h to obtain CMs. Then, the CM were centrifuged at 3000 rpm for 5 min, before the supernatant was collected and filtered with a 0.22-µm filter and immediately cryopreserved at −80 °C. The conditioned medium was concentrated using 3 kDa (Merck KGaA, Darmstadt, Germany) filters, to a concentration of 500 μg/mL, for further use in the experiment.

### 2.2. Model of Glutamate Toxicity

The experiments were performed on the PC12 rat pheochromocytoma line. PC12 cells were cultured in DMEM supplemented with 15% FBS, 100 mg/L penicillin and streptomycin at 37 °C in a 95% humidified incubator with 5% CO_2_. To modeling glutamate toxicity, cells were seeded in wells of 96 and 48 well plates, in the amounts of 10 × 10^3^ and 30 × 10^3^ cells/well, respectively. After 24 h, GPC-CM and NPC-CM were added to the culture medium, totaling a final concentration of total protein of 20 μg/mL. N-acetylcysteine (NAC, Sigma-Aldrich, St. Louis, MO, USA), measuring 750 μM, was used as a positive control. After 12 h, glutamate was added, to a concentration of 30 mM. The viability assessment and counting of the number of apoptotic cells was performed 24 h after glutamate application.

### 2.3. Cell Viability and Apoptosis Level Assays

Cell viability was measured by 2,5-diphenyl-2H-tetrazolium bromide assay (MTT, Sigma-Aldrich, St. Louis, MO, USA). Briefly, 10 μL of MTT solution (5 mg/mL) were added to each well of the 96-well plates, for an additional 2 h of incubation. The MTT reagent was then replaced with dimethyl sulfoxide (DMSO, 100 μL/well) carefully, to dissolve the formazan crystals. The absorbance at 570 and 620 nm were measured using a microplate reader (PerkinElmer, Waltham, MA, USA). The results were expressed as the percentage of the absorbance of the control cells, which was considered to be 100%. Apoptotic or necrotic cell death was characterized using Hoechst 33342 (Thermo Fisher Scientific, Waltham, MA, USA) and propidium iodide (PI, Thermo Fisher Scientific, USA) double staining. PC12 cells with different treatments were stained with Hoechst 33342 (10 mg/L) and PI (25 μg/mL) for 15 min at 37 °C. After washing with D-Hanks’ solution, the cells were observed and imaged with a fluorescence microscope (Carl Zeiss, Oberkochen, Germany) at 380/555 nm excitation and with a 535/645 nm emission for Hoechst 33342/PI respectively. Five random microphotographs were captured in each sample in different sections, and ImageJ (version 1.54f) with the Cell Counter plugin was used to analyze the apoptotic cell count.

### 2.4. Differentiation of Rat Pheochromocytoma (PC12) Cells and the Quantification of Neurite Growth

The protocol described by Rendong Hu et al. was used as the basis [32]. PC12 cells were cultured in poly-DL-lysine, and, then, 24-well plates (2 × 10^3^ cells/cm^2^) were coated in DMEM supplemented with 15% FBS, 100 U/mL penicillin and 100 mg/l streptomycin. At 12 h after plating, the medium was replaced with Opti-MEM (Gibco, Carlsbad, CA, USA), containing 0.5% FBS, GPC-CM and NPC-CM with a 20 µg/mL concentration of total protein, and 50 ng/mL NGF for the positive control group. The medium was replaced after one day. The results were collected 2 and 6 days after adding the CM. The number of neuron-like cells and the length of the neurites were counted according to the protocol described by Narayan Chaurasiya et al. [33]. The results were calculated using 100 cells in fifteen independent fields of each image. Photographs were taken on a Zeiss Axiovert microscope at 200× magnification (Carl Zeiss, Oberkochen, Germany). The quantity of neuron-like cells was calculated as (%) neurite-forming cells/total cells × 100. The length of neurites was quantified by manual tracing using ImageJ with the plugin NeuronJ.

### 2.5. Dichlorofluorescein Assay for Reactive Oxygen Species

The production of ROS was estimated with the fluorescent dye 2′,7′-dichlorodihydrofluorescein diacetate (H2DCFDA, Sigma-Aldrich, USA). H2DCFDA is non-fluorescent, but in the presence of intracellular ROS it is oxidized to highly fluorescent dichlorofluorescein (DCF). The cellular ROS was quantified using the same procedure as Hao Hong et al. [34]. Briefly, the cells on the poly-DL-lysine-coated 96-well plate were incubated with 100 μM H2DCFDA in the loading medium in 5% CO_2_/95% at 37 °C for 30 min. After the H2DCFDA was removed, the cells were washed once with DMEM and incubated for 30 min in DMEM containing GPC-CM and NPC-CM, with N-acetylcysteine used as the control. Then, 30 mM glutamate was added. After 30 min of incubation, the fluorescence per well was quantified using a plate reader (PerkinElmer, Waltham, MA, USA). The percentage increase in fluorescence per well was calculated by the following formula: scavenging rate (%) = (ft30 − ft0)/ft0 × 100%, where ft0 refers to the fluorescence at the time of 0 min and ft30 is fluorescence at the time of 30 min in the presence of glutamate.

### 2.6. Immunocytochemistry

Immunocytochemical staining for neuronal and astroglial markers was performed to confirm the phenotype of the obtained cultures. Cells were fixed with 4% formaldehyde solution (Sigma-Aldrich, USA) for 10 min, washed with phosphate-buffered saline (PanEco, Russia), incubated in phosphate-buffered saline with 0.25% Triton X-100 (Sigma-Aldrich, USA) and 1% bovine serum albumin (BSA, PanEco, Russia) for 30 min, and treated with the following primary antibodies overnight at +4 °C: anti-TUBB3 (1:800; Abcam, Cambridge, UK, ab78078), anti-S100b (1:500, Abcam, ab218515) and anti-Ki-67 (1:400, Abcam, ab15580). Then, the cells were washed with PBS and incubated with secondary antibodies, labeled with fluorochrome Alexa Fluor 555 goat anti-mouse (1:600; Invitrogen) and Alexa Fluor 488 goat anti-rabbit (1:600; Invitrogen) in the dark, for 60 min. The nuclei were stained with a solution of 1 μg/mL DAPI (4,6-diamino-2-phenylindole dihydrochloride) in phosphate-buffered saline. Photographs were taken on a Zeiss Axiovert microscope at 200× magnification (Carl Zeiss, Oberkochen, Germany).

### 2.7. Western Blot

The cells samples were lysed with a Protein Solubilization Buffer Kit (Bio-Rad, Hercules, CA, USA) and mixed with a protease and phosphatase inhibitor cocktail (Thermo Fisher Scientific, USA). Then, the supernatants were pooled, and the protein concentration was measured by the Bradford method. Laemmli Sample Buffer 2× (Bio-Rad), containing 5% β-mercaptoethanol (Bio-Rad), was added to the obtained samples. Samples were incubated at 99 °C for 10 min before performing Western blotting. The proteins were separated by sodium dodecyl-sulfate polyacrylamide gel electrophoresis (SDS-PAGE), using the TGX Stain-Free FastCast Acrylamide Starter Kit 12% (Bio-Rad), for neuritogenesis markers analysis. Protein transfer was carried out on nitrocellulose membranes using the Trans-Blot Turbo Transfer System (Bio-Rad). The membranes were pre-blocked overnight with 3% BSA in PBS and incubated with primary antibodies to β3-tubbulin (1:1000, ab7751), MAP2 (1:1000, ab32454) and β-actin (1:1000, 8457). After this, the membranes were washed and incubated with anti-rabbit IgG (H + L) cross-adsorbed secondary antibody Alexa Fluor 488 (1:1000, A-11008) or anti-mouse IgG (H + L) cross-adsorbed secondary antibody Alexa Fluor 555 (1:1000, A-21422) (Thermo Fisher Scientific) for 60 min in the dark. The immunoblots were documented using the BioRad Chemidoc Imaging System (Biorad, USA). The semi-quantitation calculation of proteins was carried out in the ImageLab 5.0 software (Bio-Rad, USA) with normalization to β-actin.

### 2.8. Quantitative Real-Time Polymerase Chain Reaction

Total cellular RNA was extracted using RNeasy Mini Kit, according to the attached instructions. Single-stranded cDNA synthesis was performed in accordance with the standard Fermentas kit protocol. For real-time PCR, the prepared reaction mixture was used with the Sybr Green intercalating dye; amplification was performed using a BioRad iQ cycler (Bio-Rad, Hercules, CA, USA). The reaction pattern was as follows: primary denaturation at 95 °C for 5 min; denaturation at 95 °C for 20 s; primer annealing at 55–64 °C for 20 s; elongation at 72 °C for 20 s (40 cycles). The Ct value was determined automatically by the accompanying software. The relative amount of mRNA was calculated using the ΔC(T) method, using Gapdh as a reference housekeeping control. The melting curve was run for each set of primers. The primers were either designed with the Primer Designing Tool (available at https://www.ncbi.nlm.nih.gov/, accessed on 10 September 2023) or their sequence is available elsewhere. The sequences and annealing temperatures of the PCR primers are presented in Table 1.

### 2.9. Statistical Analysis

The data were analyzed using a one-way analysis of variance (ANOVA) followed by a post hoc Tukey’s test using SigmaPlot 14.0 software. All data were presented as the mean ± standard deviation. *p* < 0.05 was considered to indicate a statistically significant difference.

## 3. Results

### 3.1. Characterization of Obtained Cell Cultures

The neuronal and glial progenitors were subjected to morphological analysis. The neuronal progenitors exhibited a characteristic morphology characterized by small cell bodies and neurites that did not extend beyond three cell diameters (Figure 1a). The immunocytochemistry revealed the robust expression of the neuronal marker β-III-tubulin (TUBB3) in NPCs. In contrast, glial precursors presented a spindle-shaped morphology with prominent and irregular contours, together with large oval nuclei. GPCs showed particularly high levels of expression of the astrocyte marker S100b (Figure 1b).

### 3.2. Cell Vitality and Apoptosis Level

Cell viability was examined using an MTT assay to determine the protective effect of CM on 30 mM glutamate-induced oxidative stress. CM and NAC were added to the culture medium for 12 h before glutamate treatment. Compared with the normal control group, the viability of the cells exposed to the 24 h glutamate treatment was reduced to 60.97 ± 4.01%. Pre-treatment with GPC-CM significantly attenuated glutamate-induced PC12 cell death, the viability of these cells was significantly increased to 82.47 ± 6.25% (*p* < 0.05). NPC-CM have a trend toward an increase in relative viability, with an average of 67.6 ± 5.69%. NAC substantially eliminated the negative effects of high concentrations of glutamate, with the viability being 91.34 ± 3.44% (Figure 2c). Apoptosis was measured by changing the morphology of cell nuclei, as shown in Figure 2a; apoptotic cells became thinner with pyknotic nuclei and exhibited light blue fluorescence following exposure to glutamate. The number of cells testing positive for propidium iodide in the glutamate group tended to increase, compared to the control. However, no significant differences were found between the treatment groups. The addition of glutamate increased the number of apoptotic cells from 1.82 ± 0.32% to 11.62 ± 6.25%. The number of apoptotic cells was reduced from 11.62 ± 6.25% to 8.24 ± 1.12% and 3.69 ± 0.71% for NPC-CM and GPC-CM, respectively, after 12 h of preincubation with CM (total protein of 20 μg/mL) (Figure 2b). The real-time PCR analysis revealed that exposure to glutamate increased the mRNA level of *Bax* and decreased the mRNA level of *Bcl2*. We showed that the expression of *Bcl2* and *Bax* genes varied significantly in the CM groups, which was associated with an anti-apoptotic effect. The level of *Bcl2* increased by 6.77 and 7.51 times, and *Bax* decreased by 7.32 and 25.55 times in the groups receiving NPC-CM and GPC-CM, respectively (Figure 2d,e). Micrographs of cell cultures under the influence of added substances are shown in Figure 3a. To avoid effects on proliferation, the cells were stained with the proliferation marker ki-67. Our results show that there was no difference between the groups with and without CM addition; the number of positive cells was 94.62 ± 6.73%, 97.51 ± 6.75% and 95.28 ± 5.46% for the control group, NPC-CM and GPC-CM, respectively (Figure 3b,c).

### 3.3. Levels of Reactive Oxygen Species and NFE2L2 and HMOX1 Gene Expression

We investigated the antioxidant effect of CM on PC12 cells undergoing a high-dose glutamate treatment. We found that the glutamate treatment resulted in relatively high free radical accumulation in treated PC12 cells, whereas the GPC-CM pretreatment resulted in an 11.52% reduction in free radical accumulation (*p* < 0.05). Incubation with NPC-CM for 12 h did not result in significant changes, compared to the control. The use of acetylcysteine as a positive control also resulted in a significant 19.43% reduction in ROS formation (Figure 3d). To investigate the mechanisms of the antioxidant action of CM, we measured the relative expression of *NFE2L2* and *HMOX1* genes associated with cell antioxidant defense. The addition of GPC-CM significantly increased the expression of *NFE2L2* and *HMOX1* genes by 7.56 and 9.31 times, respectively. No increase in the expression of the corresponding genes was observed with the use of NPC-CM (Figure 3d,e).

### 3.4. Neuronal Differentiation of the Rat Pheochromocytoma (PC12) Cell Line

The PC12 cell line differentiates in a neuronal direction through serum deprivation and exposure to neurotrophic factors such as NGF. The process of neurite outgrowth, during differentiation at 2 and 6 days, was evaluated for individual cells grown under different experimental conditions, using light microscopy (Figure 4a) and immunostaining against βIII-tubulin (Figure 4b). Comparative analysis of these data showed that the addition of GPC-CM significantly stimulated neurite outgrowth and changed the phenotype of neuron-like cells compared to the control group but was less pronounced when compared to the addition of 50 ng/mL NGF (Figure 4a). Cells of the control group and the NPC-CM group did not have neuron-like phenotypes. With the addition of GPC-CM, changes were observed on the second day of cultivation. We showed that treatment with GPC-CM for 6 days increased the percentage of differentiated cells from 5.23% to 42.29% (*p* < 0.01) (Figure 4c). The average neurite length of these cells increased from 38.87 μm to 93.77 μm (Figure 4d).

The relative expression of GAP43, SYN1, TUBB3 and MAP2 genes associated with neurite outgrowths was investigated. NPC-CM increased the expression of GAP43 by 1.75 times in PC12 cells. GPC-CM positively affected the expression of GAP43 (Figure 5a), TUBB3 (Figure 4b), SYN1 (Figure 4c) and MAP2 (Figure 4d) by 2.03, 1.64, 1.57 and 3.32 times, respectively. Treatment with GPC-CM led to a notable 2.37-fold elevation in MAP2 protein concentration, when compared to the control group. Although there was a tendency for an increase in TUBB3 protein levels in the GPC-CM group, no statistically significant differences were detected between the groups.

## 4. Discussion

Glutamate cytotoxicity is a pathological process associated with many neurological diseases and psychiatric disorders. The overactivation of glutamate receptors generates high levels of intracellular Ca^2+^, followed by the generation of ROS and disruption of cellular energy production. The resulting depletion of adenosine triphosphate (ATP) leads to a collapse of transmembrane electrochemical gradients, the loss of neuronal function, cell damage and death [35,36,37].

The results of our study offer valuable insights into the protective and regenerative properties of CM derived from GPCs and NPCs, in the context of glutamate-induced oxidative stress in PC12 cells. Treatment with NPC-CM showed some benefits in reducing the level of apoptosis, with the expression of *Bax* and *Bcl2* mRNAs playing a regulatory role in the processes of apoptosis, resulting in a moderate effect on viability. GPC-CM showed greater anti-apoptotic activity and effects on survival, compared with NPC-CM. In addition, GPC-CM, but not NPC-CM, demonstrated an antioxidant effect on reducing ROS levels. This antioxidant effect was substantiated by the upregulation of *NFE2L2* and *HMOX1* gene expression, highlighting the involvement of antioxidant defense mechanisms in GPC-CM’s action. Intriguingly, our study extended to neuronal differentiation, where GPC-CM demonstrated the ability to stimulate neurite outgrowth and induce a neuron-like phenotype in PC12 cells. While not as potent as the effects induced by NGF, GPC-CM initiated changes as early as the second day of cultivation, leading to a significant increase in the number of differentiated cells and the growth of neurites. These observations were confirmed by the enhanced expression of the neurite-associated genes *GAP43, SYN1*, *MAP2* and *TUBB3*, in response to GPC-CM treatment. We found marked differences in MAP2 protein levels between the groups, in contrast to TUBB3. This may be due to the fact that MAP2 protein is the predominant cytoskeletal regulator in neuronal dendrites [38]. In conclusion, although NPC-CM treatment showed some benefits in vitro in the PC12 cell line, GPC-CM, at the same concentration, had a more pronounced effect across all biological processes tested.

In a study using a rat model of diabetes, conditioned MSCs had an ameliorating effect on cognitive function and anxiety behavior. Moreover, MSC-CM treatment reduced oxidative stress by decreasing MDA and increasing GSH and antioxidant enzyme activity, as well as by decreasing TNF-α gene expression and increasing *Bcl2* gene expression in the brain [39]. A study using a rat model of a prenatal brain injury showed that the intraperitoneal administration of NSC-CM improved motor function and reduced apoptosis and inflammation, but that the effect of NSC therapy was greater [40]. Human adipose tissue MSC-CM demonstrated protective activity on the ultraviolet B-irradiated human keratinocyte cell line HaCaTs and normal human dermal fibroblasts. It was found that such CM could modulate the expression of the phase II transforming growth factor-β and heme oxygenase-1 genes, providing an anti-photoaging effect in an in vitro model [41]. Treatment with exosomes secreted by IPSC-derived MSCs demonstrated a reduction in infarct volume, an improved spontaneous walking ability, and enhanced angiogenesis through the expression of VEGF and CXCR4 proteins in a mouse model of stroke [42]. In a recent study, CM derived from dental pulp stem cells showed increased viability in a serum deprivation model and a pronounced effect on neurite outgrowth in the PC12 cell line, an effect that was more pronounced compared to a co-culture [43].

The difference in the efficacy of NPC-CM and GPC-CM appears to be due to differences in the factors secreted by the cells. In our previous study, we investigated the secretory activities of NPCs and GPCs using proteomic analysis and found that GPCs are capable of secreting large amounts of neurotrophic factors such as BDNF, NGF and GDNF. Qualitative variations in the composition of protein molecules within conditioned media were identified. Table 2 presents potential candidate proteins that may mediate differences between the performance of NPC-CM and GPC-CM [31].

Other studies have shown that NPCs secrete predominantly growth factors such as FGF and EGF [44,45]. Nevertheless, CM of various cell types contain hundreds of different protein molecules, miRNAs and exosomes and have a complex effect, which is determined by both the qualitative composition and the ratio of the substances contained in it.

## 5. Conclusions

In conclusion, our study highlights the multifaceted potential of GPC-CM as a therapeutic agent in oxidative stress-related conditions. Its protective, antioxidant, and neurotrophic effects make it a promising candidate for further research and potential applications in treating neurodegenerative disorders and injuries. Nonetheless, additional investigations are warranted to unravel the intricate mechanisms underlying these beneficial effects and to optimize the use of GPC-CM in clinical settings.

## Figures and Tables

**Figure 1 biomolecules-13-01784-f001:**
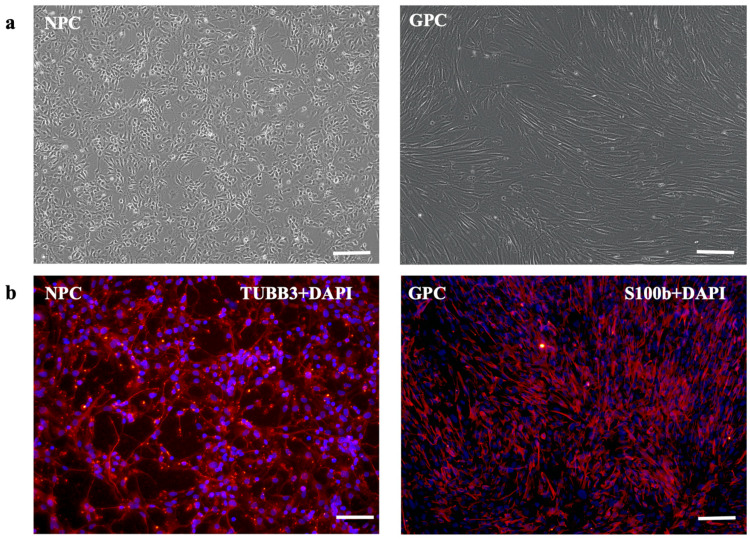
Characterization of NPCs and GPCs: phase-contrast microscopy (**a**), immunocytochemistry for S100B (glial marker) and βIII tubulin (neuronal marker) (**b**). Scale bar, 100 µm.

**Figure 2 biomolecules-13-01784-f002:**
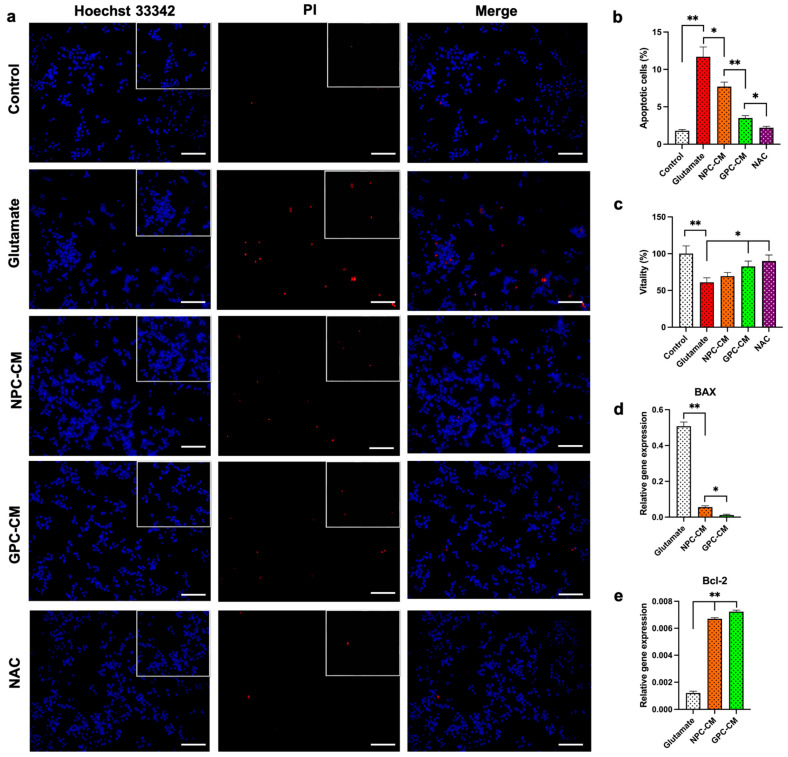
CM improve cell viability and reduce apoptosis in a model of oxidative stress induced by 30 mM glutamate for 24 h (cultured under standard conditions: 5% CO_2_ and 37 °C). Analysis of the number of apoptotic and necrotic cells, using Hoechst 33342 and propidium iodide staining in the glutamate toxicity model on the PC12 cell line: fluorescence microscopy (**a**), count of apoptotic cells (**b**). Assessment of relative cell viability using MTT assay (**c**). Relative expression of the BAX gene (pro-apoptotic marker) (**d**), and Bcl2 gene (anti-apoptotic marker) (**e**). Scale bar, 100 µm. * = *p* < 0.05, ** = *p* < 0.01.

**Figure 3 biomolecules-13-01784-f003:**
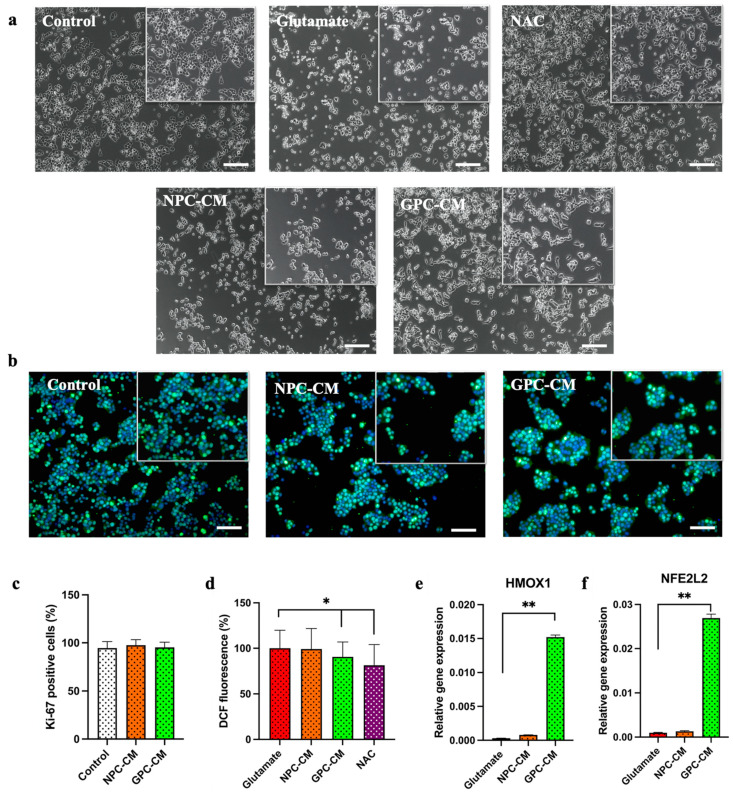
Microphotographs of glutamate-mediated cell death after 24 h obtained by phase contrast microscopy (**a**). CMs do not have significant effect on cell proliferation, as shown by immunocytochemical analysis of the proliferation marker Ki-67 (**b**) and by counting Ki-67-positive cells (**c**). GPC-CM, but not NPC-CM, have antioxidant effects on the PC12 cell line; the intracellular ROS levels are indicated by DCF-DA fluorescence intensity using a microplate reader (**d**). Relative expression of the *HMOX* gene (**e**), and *NFE2L2* gene (**f**) (antioxidant activity markers). Scale bar, 100 µm. * = *p* < 0.05, ** = *p* < 0.01.

**Figure 4 biomolecules-13-01784-f004:**
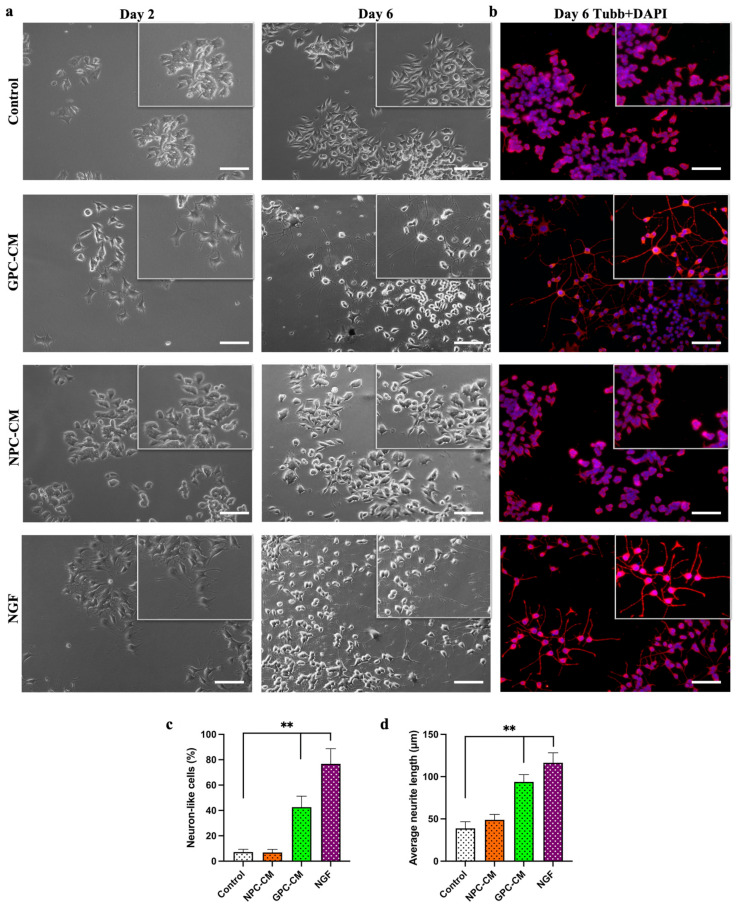
GPC-CM, but not NPC-CM, induce differentiation of PC12 cells to a neuron-like phenotype, measured using phase contrast at 2 and 6 days (**a**), and anti-tubb3 immunofluorescence staining at 6 days (**b**). Quantification of cells with a neuron-like phenotype (**c**). Neurite outgrowth was assessed by average length using ImageJ and the NeuronJ plugin (**d**). Scale bar, 100 µm. ** = *p* < 0.01.

**Figure 5 biomolecules-13-01784-f005:**
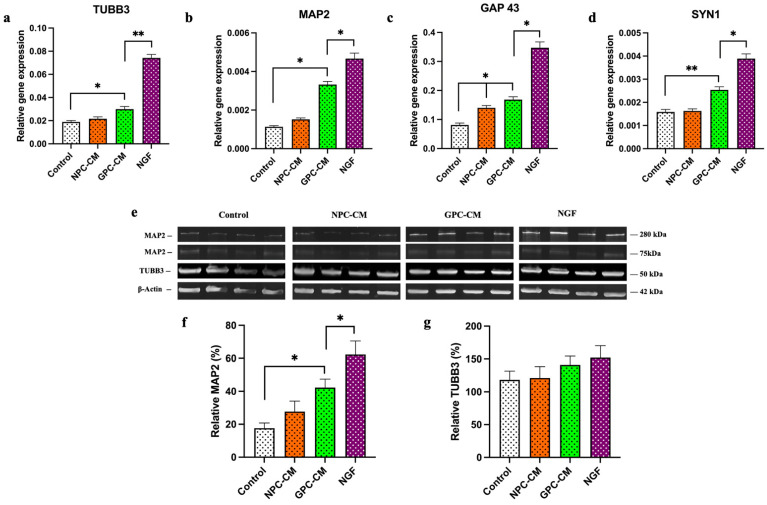
Neurotrophic effects of NPC-CM and GPC-CM assessed by qPCR and Western blot analysis. Relative expression of the TUBB3 gene (**a**), MAP2 (**b**) (neuronal differentiation marker), GAP43 gene (**c**) and SYN1 genes (**d**) (neurite outgrowth marker). Representative blots (**e**); relative protein level of the MAP2 (**f**), TUBB3 (**g**). * = *p* < 0.05, ** = *p* < 0.01. Original western blotting figures can be found in Appendix A.

**Table 1 biomolecules-13-01784-t001:** Sequences and annealing temperatures of PCR primers.

Gene	Sequence (5′ to 3′)	Annealing Temperature, °C	Product Length, bp
GAP43	for-AAGGATGATGCTCCCGTTGC rev-CGGCCTTTTCCTCTGAAGGG	64	175
βIII-tubulin	for-GGCAACTATGTGGGGGACTC rev-GCACCACTCTGACCGAAGATA	60.5	223
SYN1	for-AGCTCAACAAATCCCAGTCTCT rev-CCAGGAGAGAGGGGTTCTCA	60	249
MAP2	for-CACTTTCCGTGCCCAGATTTT rev-GCTGGTGGTATGTTCTGGCT	60	157
BAX	for-TTGTGGCTGGAGTCCTCACT rev-TTTCCCCGTTCCCCATTCATC	61	131
BCL2	for-GGGCTACGAGTGGGTACT rev-GACGGTAGCGACGAGAGAAG	55.6	148
NFE2L2	for-TGTAGATGACCATGAGTCGC rev-TCCTGCCAAACTTGCTCCAT	57.2	197
HMOX1	for-CCAGAGTTTCCGCCTCCAAC rev-CTGGGACATGCTGTCGAGC	63	179
GAPDH	for-GCGAGATCCCGCTAACATCA rev-GCTACGGGCTTGTCACTCG	62	215

**Table 2 biomolecules-13-01784-t002:** Differential comparison of secreted proteins from GPC- and NPC-derived conditioned media [31].

Biological Processes	Neuronal Progenitor Cells-Condition Medium	Glial Progenitor Cells-Condition Medium
Regulation of apoptosis and cell survival	Tissue inhibitor of metalloproteinases 2 (TIMP2) Secretogranin-2 (chromogranin C, SCG2) Wnt family member 5a (WNT5A) Neuropilin-1 (NRP1) Ras homolog family member A (RHOA) Platelet-derived growth factor D (PDGFD	Heat shock 70 kDa protein 4 (HSPA4) Heat shock protein 105 kDa (HSPH1) Hsc70-interacting protein (ST13) Leukemia inhibitory factor (LIF) Growth arrest-specific protein 6 (GAS6) Gremlin 1 (GREM1) Tetranectin (TETN)
Antioxidant protection	Catalase (CAT) Glyoxylate and hydroxypyruvate reductase (GRHPR) Peptidylglycine alpha-amidating monooxygenase (PAM)	Lysyl oxidase homolog 1 (LOXL1) Peptidylglycine alpha-amidating monooxygenase (PAM) Peroxiredoxin 4 (PRDX4) Superoxide Dismutase 1 (SOD1) Thioredoxin domain-containing protein 5 (TXNDC5) Glutathione S Transferase Omega 1 (GSTo1)
Neurogenesis and regulation of neurite growth	Ataxin 10 (ATXN10) Ephrin B1 (EFNB1) Ezrin (EZR) Fibroblast Growth Factor 8 (FGF8) Glypican 1 (GPC1) Netrin 1 (Ntn1) Neuroserpin 1 (Serpini1) Semaphorin-3C (SEMA3C) Neuronal Pentraxin II (NPTX2) Basigin (BSG) Endophilin-A2 (SH3GL1) Growth Differentiation Factor 11 (GDF11) Neurosecretory protein VGF	Dynactin (DCTN2) Thrombospondin 2 (THBS2) Prosaposin (PSAP) Sorting Nexin 3 (SNX3) Twinfilin 2 (TWF2) Cysteine and Glycine Rich Protein 1 (CSRP1) Olfactomedin Like Protein 3 (OLFML3) Ras-related Protein Rab-11A (RAB11A) Neuroblast differentiation-associated protein AHNAK

## Data Availability

All data collected or analyzed during this study are included in this article and its Appendix A.

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
