# Peer review of "Comparative Study of the Protective and Neurotrophic Effects of Neuronal and Glial Progenitor Cells-Derived Conditioned Media in a Model of Glutamate Toxicity In Vitro"

_biomolecules, 2023, doi:10.3390/biom13121784_

Round 1

Reviewer 1 Report

Comments and Suggestions for Authors

I would like to than the authors for this research. The research theme "Using cell therapy in regenerative medicine" is interesting and considered as a hot topic of research. The manuscript hypothesized that neuronal progenitor cells is better than glial cells in cell therapy due to inducing a neuroprotective and anti-oxidant properties". The conclusion is clear and the method of supporting the idea includes celluar stress and apoptotic pathways. 

I have some comments which in my experience will improve the research

1- The authors used RT-PCR for testing the mRNA expression levels. It should be tested on the protein levels either by western blot or ELISA

2- I tested the primer sequences in the primer BLAST programs, I have not found the corresponding Base Pairs length. So, kindly mention the number of base paires in Table 1

3- In the model of the glutamate toxicity, could you explain the reason of using DMEM supplement instead of RPMI 1640 that has been previously used to obtain the GPC-CM and NPC-CM in reference 31 (as mentioned) although the DMEM has lower oxygen and affects the cellular behaviour as mentioned in doi: 10.1007/s10616-009-9200-5 

Thank you for your attention 

Author Response

Authors and I very much appreciated the constructive comments on this manuscript by the reviewer. The comments have been very thorough and useful in improving the manuscript.

1. Yes, you are right. The measurement of both mRNA and protein levels would be stronger evidence and increase the validity of the study. We have added Western blot data on TUBB3 and MAP2 protein levels to the manuscript.

2. Corrected. We have added data on the length of the amplification product (some primers were selected using the “exon-exon junction” parameter). 

3. We did not use RPMI 1640 medium to obtain GPC-CM and NPC-CM. DMEM, like RPMI 1640, is widely used for the cultivation of PC-12 (doi: 10.1038/s41598-021-87431-4). In the glutamate toxicity model, DMEM was used for all experimental groups.

Reviewer 2 Report

Comments and Suggestions for Authors

Cell therapy has emerged as a pivotal strategy in addressing neurological diseases, providing potential advantages both in terms of cellular replacement and through paracrine secretory mechanisms. Yet, the methodology is not devoid of challenges, chiefly surrounding safety concerns. The deployment of conditioned stem cell media (CM) presents an innovative approach, mirroring the benefits of cell therapy but sidestepping its inherent disadvantages. In this context, the study at hand embarked on a meticulous comparative analysis of the antioxidant, neuroprotective, and neurotrophic properties of conditioned media derived from neuronal and glial progenitor cells (termed NPC-CM and GPC-CM, respectively) and their effects on the in vitro PC12 cell line. It's noteworthy that these progenitor cells were strategically differentiated from iPSCs using targeted small molecules. Impressively, GPC-CM showcased a marked reduction in apoptosis rates, while concurrently boosting cell viability, ROS levels, and upregulating antioxidant response genes, notably HMOX1 and NFE2L2, within a glutamate-induced oxidative stress paradigm. Furthermore, a distinct neurotrophic footprint was evidenced by the transformation of pheochromocytoma cell phenotype towards neuron-like structures, bolstered neurite outgrowth, and the heightened expression of pivotal genes such as GAP43, TUBB3, and SYN1. In contrast, NPC-CM administration revealed discernible antiapoptotic attributes and extended cellular longevity. Altogether, this investigation accentuates the promising therapeutic horizon of conditioned media in the rapidly evolving domain of regenerative medicine.

Major :

1. Elucidation of Regulatory Media Composition: A detailed analysis of the regulatory media's components is essential. Using techniques like proteomics or cytokine array analysis can reveal these components accurately. It's important to note that referencing methodologies from the lead author's previous study, 'Therapeutic Effects of hiPSC-Derived Glial and Neuronal Progenitor Cells-Conditioned Medium in Experimental Ischemic Stroke in Rats, Diana Salikhova et al., Int. J. Mol. Sci.' enhances the current research's validity and ensures consistency in experimental approaches.

2. Temporal and Concentration-centric Analysis: Examine the regulatory media's effects over time and across varying concentrations. Incorporating a broader range of concentration and time conditions would be beneficial. Changes based on these variables are crucial for therapeutic use. A thorough understanding of these shifts will enrich the study's depth and clarity.

Conclusive Thoughts: While the present research exemplifies a transformative intervention in the domain, there lies an opportunity to augment its academic stature. By infusing nuanced data interpretations, fostering ties with antecedent studies, and indulging in deeper analytical traverses, the investigative horizon stands promisingly broadened.

Minor opinions:

1. In Fig2, the PI signal appears considerably subdued. For consistency with the statistical graphics, I suggest either amplifying this signal or recapturing it to represent a more indicative region.

2. There seems to be a misalignment between the statistically presented expressions and the descriptive results. For instance, when examining the statement, 'The addition of glutamate increased the number of apoptotic cells from 1.82±0.32% to 11.62±6.25%. Compared with the untreated group, the number of apoptotic cells decreased to 8.24±1.12% and 3.69±0.71% after preincubation with NPC-CM and GPС-CM, respectively (figure 2c).' it appears there is a requirement for a comparative outcome between the untreated group and the groups preincubated with NPC-CM and GPС-CM on the graph. It's essential to ensure alignment and coherence between the described results and their graphical representation.

3. Reviewing Figure 3 (B), while the outcomes are interpreted as non-differential, there is an absence of supportive statistical evidence. Although the representative image seems to buttress the described results, a rigorous statistical analysis would provide more definitive clarity.

Author Response

Authors and I very much appreciated the constructive comments on this manuscript by the reviewer. The comments have been very thorough and useful in improving the manuscript.

Major :

1. In the Discussion section, we have added data on the qualitative differences detected in the protein composition of NPC-CM and GPC-CM. Potential protein molecules that may be responsible for the differences in effects between the CMs are indicated.

2. Yes, you are right. Information on the dose-dependent effect of CMs could improve the validity of the study. However, this research is a pilot study and has addressed the wide range of potential effects of NPC-CM and GPC-CM. In this study, we selected the concentration of CMs based on literature data. In future studies, we intend to examine the dose-dependent effect of CMs and to investigate the mechanisms of action in more detail.

Minor:

1. Corrected. The signal from the Pi stain is enhanced and should look more informative.
2. Corrected. The formulation of the results has been changed, and we have made changes to the graphs.
3. Corrected. We have added a graph describing the number of Ki-67 positive cells in different experimental groups.